# Evaluation and Comparison of the STIMUL Extended and Simplified Risk Scores for Predicting Two-Year Death in Patients Following ST-Segment Elevation Myocardial Infarction

**DOI:** 10.3390/medicina57121349

**Published:** 2021-12-10

**Authors:** Svitlana Korol, Agnieszka Wsol, Alexander Reshetnik, Alexander Krasyuk, Kateryna Marushchenko, Liana Puchalska

**Affiliations:** 1Department of Military Therapy of the Ukrainian Military Medical Academy, 01015 Kyiv, Ukraine; valueva.sv@gmail.com (S.K.); alexkrasyuk@ukr.net (A.K.); katipon@gmail.com (K.M.); 2Department of Experimental and Clinical Physiology, Laboratory of Centre for Preclinical Research, Medical University of Warsaw, 02-091 Warsaw, Poland; liana.puchalska@wum.edu.pl; 3Department of Nephrology and Intensive Care Medicine, Charité—Universitätsmedizin Berlin, 13353 Berlin, Germany; alexander.reshetnik@charite.de

**Keywords:** ST-segment elevation myocardial infarction, STEMI, risk, score, two-year death

## Abstract

*Background and Objectives:* The management of ST-segment elevation myocardial infarction (STEMI) requires a patient’s long-term risk to be estimated. The objective of this study was to develop extended and simplified models of two-year death risk estimation following STEMI that include and exclude cardiac troponins as prognostic factors and to compare their performance with each other. *Materials and Methods*: Extended and simplified multivariable logistic regression models were elaborated using 1103 patients with STEMI enrolled and followed up in the STIMUL (ST-segment elevation Myocardial Infarctions in Ukraine and their Lethality) registry. Results: The extended STIMUL risk score includes seven independent risk factors: age; Killip class ≥ II at admission; resuscitated cardiac arrest; non-reperfused infarct-related artery; troponin I ≥ 150.0 ng/L; diabetes mellitus; and history of congestive heart failure. The exclusion of cardiac troponin in the simplified model did not influence the predictive value of each factor. Both models divide patients into low, moderate, and high risk groups with a C-statistic of 0.89 (95% CI 0.84–0.93; *p* < 0.001) for the extended STIMUL model and a C-statistic of 0.86 (95% CI 0.83–0.99; *p* < 0.001) for the simplified model. However, the addition of the level of troponin I to the model increased its prognostic value by 10.7%. *Conclusions*: The STIMUL extended and simplified risk estimation models perform well in the prediction of two-year death risk following STEMI. The simplified version may be useful when clinicians do not know the value of cardiac troponins among the population of STEMI patients.

## 1. Introduction

Patients with ST-segment elevation myocardial infarction (STEMI) are at high risk of in-hospital and long-term adverse outcomes [1,2,3]. This risk depends on different factors, for example, medical background, clinical signs, symptoms at admission, and the management during the initial hours of acute coronary syndrome (ACS) development. To estimate the risk of long-term outcomes, these factors are included in multivariate risk score models [4,5,6,7,8,9,10,11]. The majority of these models have been derived from databases from clinical trials, which tend to exclude high-risk patients. Other risk estimation models have been developed using databases, which may be limited by their inclusion criteria, e.g., only primary coronary interventions or fibrinolytic therapy were used (e.g., TIMI-STEMI, In-TIME II, and SYNTAX) [4,5,6]. These scores might not accurately represent the spectrum of patients in clinical practice and may lead to inappropriate decision making.

Today, the GRACE risk score, which was developed as part of the multinational Global Registry of Acute Coronary Events (GRACE), is considered as one of the most accurate for determining in-hospital and long-term outcomes [7,12,13]. The GRACE risk score also has a simplified version (mini-GRACE) applied in cases when clinicians do not know a patient’s Killip class or glomerular filtration rate (GFR) at the time of admission to hospital [8].

Ukrainian cardiology centers did not take part in the GRACE registry, and validation of the GRACE risk scores has not been undertaken for Ukraine either. The first Ukrainian survey of ACS with ST-segment elevation–STIMUL (ST-segment elevation Myocardial Infarctions in Ukraine and their Lethality) [14] showed that the level of determination of cardiac troponin values was low at admission among the population of STEMI patients. This is region specific as, in Ukraine, which is a low-income European country, an evaluation of troponins frequently requires patients to pay, and lots of patients with STEMI cannot afford to pay for laboratory tests. However, the GRACE risk scores include troponins [7,8]. For these reasons, the use of the GRACE risk assessment models is limited in Ukraine. Therefore, the objective of this study was to develop extended and simplified models of two-year death risk estimation following STEMI that include and exclude cardiac troponins as prognostic factors and compare their performance with each other.

## 2. Materials and Methods

### 2.1. Study Design

Details of the STIMUL (ST-segment elevation Myocardial Infarctions in Ukraine and their Lethality; 2013) registry have been described previously [14]. In brief, 1103 patients who met the criteria of ST-segment elevation acute coronary syndromes (ACS) were enrolled in the registry at the hospital and followed up over a 24-month period in 2008–2013. This study analyzed a whole range of data on patients with ST-segment elevation ACS who were admitted to three cardiology departments (two in Khmelnytskyi and one in Vinnytsia) of the central regions of Ukraine within 24 h of the development of a cardiac incident. The enrollment of patients in the STIMUL registry did not influence the in-hospital and out-hospital medical strategy. Informed consent was obtained from all patients at the time of enrollment.

### 2.2. Statistical Analysis

All analyses were performed using SAS software (SAS Institute Inc., Cary, NC, USA). Cross-tabulation analysis based on Chi-square (χ^2^) was used to assess the correlations between variables and the event of death during a two-year period. Logistic regression was used to predict a binary outcome of the model based on χ^2^, log likelihood, Cox and Snell’s R^2^, and Nagelkerke’s R^2^. Discriminant analysis based on Wilk’s lambda made it possible to test how well each level of independent variable contributed to the model. The model was generated using stepwise logistic multivariate analysis by Cox regression. The points of each variable for the two-year death model were determined by comparison of unstandardized canonical discriminant function coefficients with standardized beta coefficients in logistic regression. Areas under ROC (receiver operating characteristic) curves were used to assess the discrimination of models. The coefficient of determination R^2^ was used to compare the performance of different risk estimation models. A *p*-value ≤ 0.05 was considered to be statistically significant.

## 3. Results

### 3.1. Baseline Characteristics of the STIMUL Population

The baseline characteristics of all patients included in the STIMUL registry are presented in Table 1. The average age among the Ukrainian population with a STEMI diagnosis at admission to hospital was 63.4 ± 11.5 years. A total of 17.8% of the patients were 75 years and older; 74.3% (*n* = 819) individuals were males; 24.2% of the patients had a past history of myocardial infarction, and 22.8% were diagnosed with heart failure. Hypertension (diagnosed in the past or de novo) was found in 76.6% of patients, while a body mass index > 30 kg/m^2^ and diabetes were present in 32.0% and 24.9% of the patients, respectively. According to the baseline data (Table 1), the STIMUL population was characterized by high or very high cardiovascular risk.

The average heart rate at admission was 83.4 ± 2.6 beats per minute; systolic blood pressure was 138.6 ± 3.6 mmHg. A total of 23.8% of the study patients (*n* = 262) were classified as Killip ≥ II at hospital admission. Cardiogenic shock was diagnosed in 39 (3.5%) cases. The mean GFR was 78.4 ± 2.3 mL/min. Average cardiac troponin I concentration (Troponin I ELISA; Biomerica, Irvine, CA, USA) at admission was 230.5 ± 7.0 ng/L. The mean level of blood glucose in the STIMUL population was 7.4 ± 1.0 mmol/L and 11.07 ± 0.4 mmol/L among patients with diabetes. The average total cholesterol value was 5.6 ± 1.4 mmol/L, and the average hemoglobin level was 144.3 ± 1.8 g/L; 4.9% of the patients had a hemoglobin level < 100.0 g/L. In addition, 34.5% (*n* = 381) of the patients had a high risk of in-hospital mortality according to their GRACE score. A total of 11.9% (*n* = 131) and 19.5% (*n* = 215) of the patients, respectively, had a high and a very high bleeding risk as assessed by the CRUSADE score.

The median time from symptom onset to admission at the emergency department was 5.1 ± 0.3 h (0.3–21.5 h). A total of 59.8% (*n* = 660) of the patients were admitted to cardiology units during the first six hours, and 71.1% of the patients (*n* = 784) during the first 12 h. However, only half of them (51.4%, 339 patients) were hospitalized in interventional departments, and 73.4% (*n* = 237) of the patients admitted to interventional units underwent primary percutaneous coronary intervention (PCI) with a door-to-balloon time of 4.5 ± 0.2 (1.2–7.8) h. Fibrinolytic therapy was performed in 8.4% (*n* = 93) of the cases. Other patients with ST-segment elevation ACS did not receive any reperfusion therapy. The major reasons for not performing reperfusion therapy were the unavailability of catheterization laboratories (40.5%; *n* = 447) and late arrival (27.8%; *n* = 307). Additional important reasons were contraindications (7.4%), uncertain diagnosis (6.3%), and patient refusal (6.1%). High rates of patient refusal and contraindications in the STIMUL registry were due to economic reasons or low awareness of the disease, and a high and a very high bleeding risk assessed by the CRUSADE score, respectively.

The in-hospital mortality rate was 11.3% (*n* = 125) in the STIMUL cohort and 7.0% (*n* = 23) among patients who underwent coronary reperfusion (*p* < 0.001). During the first 6, 12, and 24 months after STEMI, 64 (7.3%), 140 (16.1%), and 169 patients (19.4%) died, respectively (Figure 1).

The demographic characteristics, medical history, presenting clinical features, and laboratory findings were collected and analyzed.

### 3.2. Determination of Risk Score

The cross-tabulation analysis showed significant links (*p* < 0.001) between death over a two year period and the Killip class at admission (χ^2^–59.3), history of congestive heart failure (χ^2^–104.6), diabetes (χ^2^–60.7), body mass index ≥ 30 kg/m^2^ (χ^2^–39.3), elevated cholesterol level (χ^2^–36.5), family history of coronary artery disease (χ^2^–20.9), history of prior MI (χ^2^–17.8), hemoglobin level < 100 g/L (χ^2^–46.0), time from cardiac arrest to admission (χ^2^–32.6), age ≥ 75 years (χ^2^–17.1), female (χ^2^–22.2), reperfusion therapy (χ^2^–19.4) resuscitated cardiac arrest (χ^2^–21.8), triple vessel coronary artery disease (χ^2^–15.7), level of systolic blood pressure < 100 mmHg (χ^2^–14.71), heart rate ≥ 100 beats per minute (χ^2^–19.0), and glomerular filtration rate ≤ 65 mL/min (χ^2^–13.1). These variables were then assessed by the logistic regression method (Table 2).

From Table 2, it can be seen that a few of the variables are consistent, powerful predictors of the two-year risk of death among the population following STEMI, and these variables were included in the model. These variables were: age, gender, Killip class at admission, successful resuscitation of sudden cardiac death, history of congestive heart failure, diabetes, and reperfusion therapy.

The average age among men and women who died in the following two years was 65.9 ± 1.4 vs. 73.0 ± 1.1 years (*p* < 0.001). However, men who died in the following two years were 7 years younger than women. The rate of two-year death was 2.5 times higher among women: 30.7% vs. 12.3% in men (*p* < 0.001). Therefore, the mortality rate among men and women varied in different age groups. To increase the statistical significance of the model, the age category was divided into several groups, where an increased risk of death was associated with age. Due to variations in the mortality levels between males and females in different age groups, gender was not included as an independent prognostic factor in the model.

Cardiac troponin values were higher among patients who died within the following two years: 196 ± 24 ng/L in contrast with 141 ± 12 ng/L among survivors (*p* < 0.001). Troponin I ≥ 150.0 ng/L had the highest sensitivity and specificity; therefore, it was set as the threshold above which the risk of two-year death increased substantially.

Thus, seven independent predictors were included in the model of two-year death following STEMI. These were: age; Killip class ≥ 2 at admission; troponin I ≥ 150.0 ng/L; resuscitated cardiac arrest; non-reperfused infarct-related artery; diabetes; and a history of congestive heart failure (Table 3).

A certain number of points were given to each predictor (depending on its prognostic weight) based on the odds ratio and its confidence intervals. As a result, the two-year mortality risk estimation model was elaborated (Patent No. 83741 Ukraine) (Table 3).

The sum of the points defined the risk of patients with STEMI and divided them into three groups of low (from 0 to 4.5 points), moderate (5.0–10.0 points), and high risk (10.5 points or more) of two-year death (Table 4). The distribution of death rates with the number of points in the STIMUL extended risk score model among patients with STEMI is presented in Figure 2.

The results of the derivation (Table 5) of the two-year death risk model are presented below.

The performance of the extension model of long-term adverse events estimation among patients with STEMI was high (*p* < 0.001) with a C-statistic of 0.89 (95% CI 0.84–0.93; *p* < 0.001) and the prediction value of two-year death in 51.8% of the cases (OR 1.8; 95% CI: 1.5–2.1; *p* < 0.001). The ROC curve for the extended risk score of two-year death after STEMI is presented in Figure 3.

Cardiac troponin values were determined at admission in only 17.1% of the patients in the STIMUL registry. Therefore, the model presented above may not be applicable for the majority of patients with STEMI in Ukraine. This also suggests the need for the establishment of a mortality risk model without cardiac troponin value estimation. For this reason, troponin was excluded from the STIMUL risk prediction model. The simplified STIMUL model for estimating the risk of two-year death was proposed (Patent No. 83744, Ukraine) and is presented in Table 6.

Table 6 shows that no variable changes their predictive weight in the simplified STIMUL risk score in comparison with the extended STIMUL risk score. Summing the points of every prognostic factor makes it possible to define the risk of two-year death among patients with STEMI and divide them into categories of low, moderate, and high risk of death (Table 7). The distribution of death rates with the number of points in the STIMUL simplified risk score model among patients with STEMI is presented in Figure 4.

However, in comparison with the extended model, the number of points that defines the risk of two-year death decreases by 0.5 points for each level of risk. The results of testing the derivation of the risk score of two-year death following STEMI among the patient population without the determination of cardiac troponins at admission were estimated by the logistic regression method and are presented in Table 8.

The prognostic value of the two-year death model among the patient population following STEMI without cardiac troponin evaluation at admission remains significant (*p* < 0.001). However, the performance of the two-year death model was 41.1% (OR 1.7; 95% CI: 1.6–1.9; *p* < 0.001) versus 51.8% (OR 1.8; 95%: 1.5–2.1; *p* < 0.001) in the extended score, which included cardiac troponin values, which shows that the addition of the level of troponin I to the model increases its prognostic value by 10.7%. The prediction level of survival was 95.9%. The ROC curve for the simplified risk score of two-year death following STEMI among the patient population whose cardiac troponins were not determined is presented in Figure 5.

The ROC curve in Figure 5 does not cross the diagonal line, representing an uninformative test. The larger area under the curve indicates the high prognostic significance of the model (C-statistic 0.86; 95% CI 0.83–0.89, *p* < 0.001).

To summarize, this simplified model could be applicable for patients who did not undergo tests to determine troponin levels at admission. However, the determination of cardio-specific markers increases its prediction value of two-year death to 51.8%.

The coefficient of determination R^2^ was used to compare the performance of the two-year mortality risk estimation by STIMUL and GRACE scores [7,8]. The coefficient was 75.0% and showed that the two risk prediction models are similar.

## 4. Discussion

Several risk estimation models have been developed to predict long-term prognosis among patients with STEMI. These are GRACE [7,8], PAMI (Primary Angioplasty in Myocardial Infarction) [9], and CADDILAC (Controlled Abciximab and Device Investigation to Lower Late Angioplasty Complications) [11]. The GRACE score is considered to be the most accurate for the population with ACS. It has been validated using numerous databases [12,13,15,16,17,18,19,20] and is currently recommended by the NICE guidelines for risk stratification of patients with ACS [21].

Seven independent prognostic factors were used for the population with STEMI included in the STIMUL registry: age; Killip class ≥ 2 at admission; successful resuscitated cardiac arrest; troponin I ≥ 150 ng/L; non-reperfused infarct-related artery; diabetes mellitus; and a history of congestive heart failure.

To increase the statistical significance of the model, the age category was divided into several groups, with increasing risk of death. The age-oriented division was also used in other estimation risk models [7,8,9,11] that predict long-term outcomes. As in other models [7,8,9,11], gender was not included in the STIMUL risk score as a separate independent prognostic factor because the two-year death risk among males and females varied in different age groups. However, in a short-term observation, Cenko et al. reported that women with obstructive coronary artery disease were at a higher risk of mortality [22].

The presence of Killip class at admission appeared to be a considerable prognostic factor. Killip class at admission was also a recognized prognostic factor in GRACE, PAMI, and CADDILAC risk scores [7,8,9,11]. Resuscitated cardiac arrest was used as a predictor of long-term prognosis for the first time. Non-reperfused infarct-related artery was revealed as an important factor of the risk of two-year death. This explains why reperfusion therapy is indicated for all patients with symptoms of ischemia and persistent ST-segment elevation and has the highest level of evidence (I A) in the European STEMI management guidelines [1]. In the PAMI and CADDILAC models [9,11], the risk of one-year death also depends on the grade of coronary flow by TIMI (the Thrombolysis in Myocardial Infarction Trial) [23].

A hemoglobin level < 100 g/L during STEMI increased the risk of two-year death by 6.2 times (95% CI 3.4–11.2; *p* < 0.001) in the population of Ukrainian STEMI patients. This variable was also used as an independent prognostic factor in the CADDILAC risk score [11]. However, in the STIMUL risk score, it appeared to be statistically insignificant (OR 0.4; 95% CI 0.1–3.0; *p* > 0.005).

It was similar for a GFR ≤ 65 mL/min. This variable was an independent prognostic factor for the STIMUL risk score of in-hospital death [24]. GFR was also included in the GRACE score and determines the in-hospital and long-term prognosis [7,8]. The criterion “GFR ≤ 65 mL/min” increased the risk of two-year mortality by 2.4 times among patients enrolled in the STIMUL registry (95% CI 1.5–3.9; *p* < 0.001). However, after calculating the long-term prediction model, it appeared to be statistically insignificant.

The risk of two-year death also increases among patients with congestive heart failure and diabetes before STEMI. Congestive heart failure was also included as an independent risk factor in the GRACE and CADDILAC risk scores [7,11], while diabetes was used in the PAMI risk score for prediction of 6-month death among patients with STEMI [9].

The cardiac troponin values were higher among patients who died during the following two years than among those who managed to survive (*p* < 0.001). The troponin I level ≥ 150 ng/L had the highest sensitivity and specificity and was defined as the threshold level for the risk of two-year death. This criterion was also included in the GRACE risk scores [7,8].

The number of cardiac troponin assessments was quite small among the population of STEMI patients in Ukraine due to economic aspects [14]. This explains the exclusion of this criterion from the presented simplified STIMUL risk score model. For similar reasons, the mini GRACE score was proposed [8] as it makes it possible to estimate the risk among individuals when the Killip class or creatinine values are unknown. However, the evaluation of troponins in all cases of STEMI patients would make it possible to define the risk of long-term adverse outcomes for a larger portion of the STEMI population.

Excluding cardiac troponins from the model did not influence the value of each of the independent prognostic factors. The model makes it possible to divide patients into three groups of low, moderate, and high risk of two-year death. However, the amount of points reflecting a certain level of risk decreased by 0.5.

In the extended STIMUL risk score, the validity of the model was calculated with the possibility of two-year death prediction in 51.8% of cases. The model remained significant with a C-statistic of 0.89 (95% CI 0.84–0.93; *p* < 0.001).

The STIMUL simplified risk predicting model also shows good discriminatory performance for mortality at up to two years following discharge from hospital with STEMI with a C-statistic of 0.86 (95% CI 0.83–0.89; *p* < 0.001). In contrast to the STIMUL risk estimation models, the C-statistic for the GRACE and CADDILAC scores was 0.8 [7,11]. The data obtained proved the high prognostic capability of the model.

In our study, the rate of reperfusion therapy in Ukrainian patients with STEMI was dramatically low. Limited access to PCI procedures at the time of registration (2008–2013) and late hospitalizations were the main barriers to invasive reperfusion treatment and had a great impact on in-hospital mortality, which was 11.3% [14]. Similar to our results, Kämpfer et al. have shown a low rate of reperfusion therapy in ACS in Ukraine [25]. Namely, in their study, significant differences in the number of primary PCI procedures for ACS in 2010 between three different socioeconomic environments (Switzerland, Poland, and Ukraine) were reported. In Switzerland and Poland, coronary interventions were the first choice therapy, whereas in Ukraine, only 30% of patients with ACS received fibrinolysis therapy, and primary PCI was not performed at all as it was unavailable at this time in the study centers in Ukraine. Our results also correspond with the Euro Heart Survey 2009 Snapshot where, in comparison with Western European countries, the use of PCI for STEMI treatment in Eastern European countries was low (23%), and 44% of patients that called within 12 h of symptom onset did not receive reperfusion therapy [26]. High rates of contraindications and patient refusal in the STIMUL registry were due to economic reasons or low awareness of the disease, and a high and a very high bleeding risk assessed by the CRUSADE score. In the Second Euro Heart Survey on ACS, the prevalence of contraindications was similar to that of our study (6.5%), as well as uncertain diagnosis (11.2%) and late arrival (30.1%) [27].

In conclusion, for many of the patients enrolled in the STIMUL registry who were admitted to cardiology units within the therapeutic window, PCI procedures were not available at the time due to a lack of an organized PCI network in Ukraine. During the STIMUL registry, there was only one catheterization laboratory (Cath Lab) in Khmelnytskyi and one in Vinnytsia, which was without 24/7 availability at the time. The other problem was that patients could not afford to buy stents when the European Cardiac Society “Stent for Life” initiative was launched in Ukraine in September 2011, and in 2013, the PCI became available for 30% of the Ukrainian population. Currently, according to the Ukrainian Ministry of Health reports, the total number of Cath Labs is 42, whereas the population of Ukraine is over 44 million (1 Cath Lab for 1,048,000 people) [28]. In Switzerland, there is one Cath Lab for 230,000 people and in Poland, a country neighboring Ukraine with a population of 37.97 million, there is one Cath Lab for 239,000 people. In our study, we showed that a lack of infarct-related artery reperfusion in STEMI patients increased the risk of death by almost six-fold during the two years following myocardial infarction. Therefore, the results of the STIMUL registry revealed the immense need for improvement of cardiac care for patients with acute coronary syndromes in Ukraine by increasing accessibility to invasive reperfusion procedures, but also specialist healthcare.

## 5. Conclusions

The extended STIMUL risk estimation model of two-year death includes the following prognostic factors: age, Killip ≥ 2 at admission, non-reperfusion of IRA, resuscitated cardiac arrest, troponin I level ≥ 150 ng/L, diabetes, and a history of congestive heart failure. The model divides the population of STEMI patients into three groups of low, moderate, and high risk.The simplified STIMUL risk score includes the same prognostic factors as the extended STIMUL risk score, apart from cardiac troponin assessment. The exclusion of troponins did not influence the prognostic weight of each predictor. The model also divides patients into three groups of risk.Both STIMUL risk prediction models of two-year death following STEMI perform well with a C-statistic of 0.89 (95% CI 0.84–0.93; *p* < 0.001) for the extended model and a C-statistic of 0.86 (95% CI 0.83–0.99; *p* < 0.001) for the simplified model. However, exclusion of troponins from the simplified model decreased its prognostic value by 10.7%.

## 6. New and Noteworthy

STIMUL is the first registry in Ukraine with long-term follow-up of patients. Ukraine has not taken part in long-term surveys, such as GRACE, and there are no validations of the models based on these studies. For this reason, such models are not applicable for the Ukrainian population of STEMI patients.

The data from the Ukrainian survey revealed the low level of troponin assessment at admission. This limits the use of the majority of well-recognized models for such patients. Data from the STIMUL survey make it possible to elaborate the extended and simplified risk prediction scores for patients whose troponin levels were not determined.

## 7. Perspectives

Both the extended and the simplified STIMUL models for two-year death risk estimation require validation in other populations to provide a reliable assessment of its utility. A detailed analysis comparing the GRACE and STIMUL risk scores with and without troponins for the prognosis of the risk of death should be undertaken.

## 8. Limitations of the Study

The extended and simplified risk stratification models of two-year mortality among the population of STEMI patients were validated only in the Ukrainian STEMI population and need to be validated in other populations of STEMI patients to confirm their prognostic value. In particular, their validation and comparison with STEMI populations of medium-income and high-income countries are necessary. It should be noted that in the STIMUL study, the number of revascularizations (PCI and fibrinolysis) was low due to limited access to invasive reperfusion procedures in ACS, the low number of specialist healthcare units in Ukraine, and the low awareness of the population concerning this disease. This resulted in a relatively high incidence of non-reperfused infarct-related artery, a prognostic factor that was included in both the extended and the simplified STIMUL risk score models.

## 9. Patents

Both extended and simplified STIMUL risk scores were patented with the respective patents numbers:
Patent No. 83741, Ukraine; https://uapatents.com/8-83741-sposib-prognozuvannya-dvorichno-smertnosti-pislya-gostrogo-infarktu-miokarda-z-zubcem-q.html (accessed on 8 October 2021)Patent No. 83744, Ukraine; https://uapatents.com/8-83744-sposib-sproshheno-ocinki-riziku-nespriyatlivogo-viddalenogo-naslidku-gostrogo-infarktu-miokarda-z-zubcem-q.html (accessed on 8 October 2021).

## Figures and Tables

**Figure 1 medicina-57-01349-f001:**
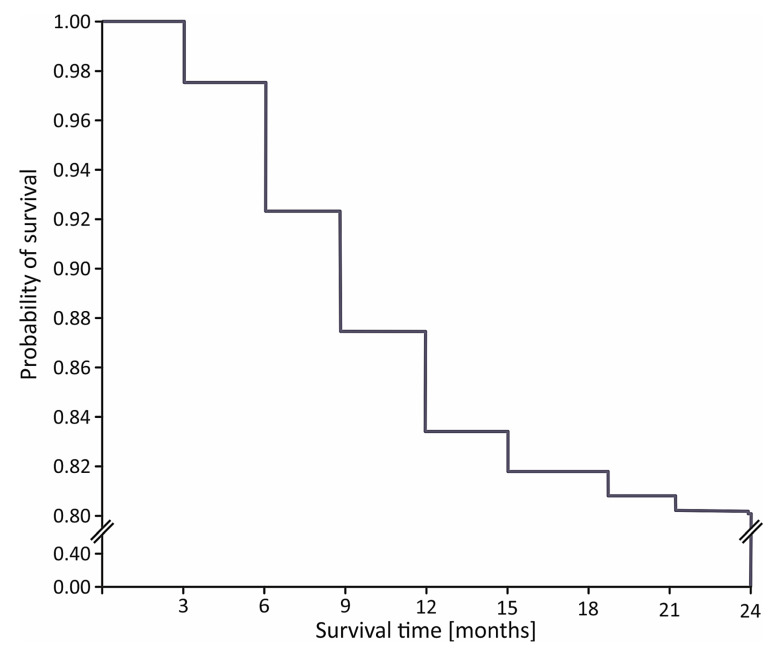
Kaplan–Meier curve for 24-month survival in STIMUL registry population.

**Figure 2 medicina-57-01349-f002:**
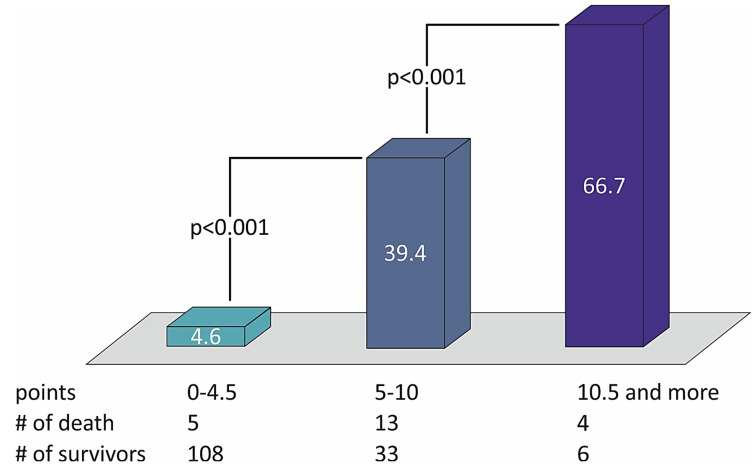
The distribution of death rates with the number of points in the STIMUL extended risk score model among patients with STEMI.

**Figure 3 medicina-57-01349-f003:**
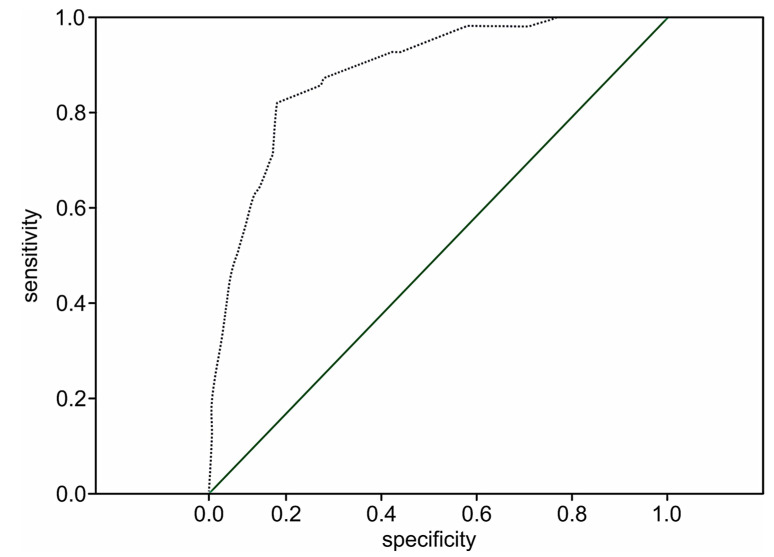
The ROC curve for the extended STIMUL risk estimation model of two-year death following STEMI.

**Figure 4 medicina-57-01349-f004:**
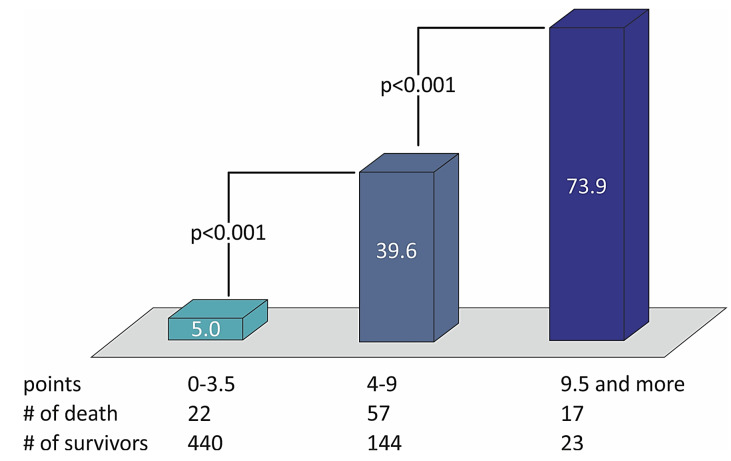
The distribution of death rates with the number of points in the STIMUL simplified risk score model among patients with STEMI.

**Figure 5 medicina-57-01349-f005:**
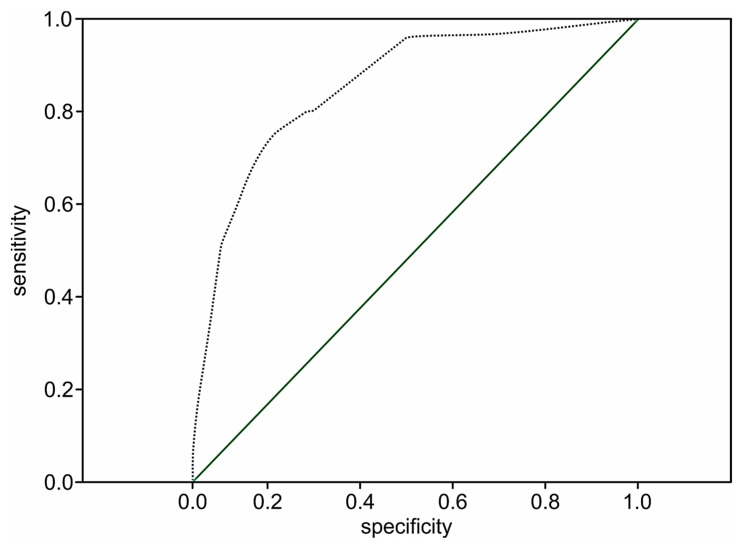
The ROC curve for the simplified STIMUL prediction model of two-year death risk following STEMI.

**Table 1 medicina-57-01349-t001:** Baseline, demographic, and clinical characteristics of the STIMUL registry population.

Characteristic	*N*	%
Age, years	63.4 ± 11.5	
Male gender	819	74.3%
Hypertension	845	76.6%
Hyperlipidemia (defined as total cholesterol ≥ 4.5 mmol/L)	565	50.7%
Body mass index (BMI) > 30 kg/m^2^	353	32.0%
Family history of coronary artery disease	351	31.8%
Diabetes mellitus	275	24.9%
Current smoker	300	27.2%
Past smoker	354	32.1%
Prior angina	380	34.5%
Prior myocardial infarction	267	24.2%
Prior percutaneous coronary intervention	23	2.1%
Prior coronary bypass graft surgery	3	0.3%
Prior heart failure	251	22.8%
Prior stroke/transient ischemic attack	72	6.5%
Prior renal failure	19	1.7%
Heart rate, mean bpm	83.4 ± 2.6	
Systolic blood pressure, mmHg	138.6 ± 3.6	
Killip class, ≥II - cardiogenic shock	26 239	23.8% 3.5%

**Table 2 medicina-57-01349-t002:** Predictors of death over a two year period among the population of STEMI patients.

Factor	B Coefficient	Standard Deviation	Wald	*p*	OR	95% CI	
						Low	High
Age ≥ 75	0.04	0.02	4.3	<0.05	1.0	1.0	1.1
Female	0.7	0.5	1.9	<0.05	1.3	1.2	1.4
Killip ≥ II	3.3	0.5	47.7	<0.001	27.7	10.9	72.0
Resuscitated cardiac arrest	4.2	1.3	10.6	<0.001	63.2	5.2	76.9
No reperfusion therapy	1.7	0.3	27.7	<0.001	5.7	3.0	10.8
Diabetes	1.2	0.4	8.0	<0.001	3.3	1.4	7.5
Heart failure	2.1	0.4	25.9	<0.001	8.0	3.6	17.9
Constant	6.1	1.3	21.8	<0.001	0.02		

**Table 3 medicina-57-01349-t003:** STIMUL risk score for prediction of two-year death among the patient population with ST-segment elevation myocardial infarction. ACS: acute coronary syndrome.

Independent Prognostic Factor	OR	95% CI		The Ratio of the Scope	Mortality Level %	Points
		Low	High			
Age, years						
˂40	–	–	–	–	<1	0.5
40–59	1.8	1.2	1.9	1.6	21.0	1.0
60–69	2.1	1.2	2.7	2.3	26.0	2.5
≥70	3.7	2.1	4.8	2.3	52.0	3.5
Killip class						
0–I	–	–	–	–	–	0
≥II	15.5	8.6	28.1	3.3	82.0	4.0
Reperfusion						
yes	–	–	–	–	–	0
no	5.7	3.0	10.8	3.6	17.0	2.0
Troponin I ≥ 150.0 ng/L	2.1	1.7	5.1	3.0	46.0	2.0
Resuscitated cardiac arrest	8.3	7.0	12.3	1.8	7.0	2.0
Diabetes mellitus	3.0	1.9	4.9	2.6	43.0	2.5
History of congestive heart failure prior to ACS	4.0	2.5	6.3	2.5	50.2	3.5

**Table 4 medicina-57-01349-t004:** The number of points for two-year cardiovascular death risk estimation in the STIMUL risk score model among the population of STEMI patients.

Points	Risk %	Risk
**0–4.5**	5.0	LOW
**5.0–10.0**	40.0	MODERATE
**≥10.5**	70.0	HIGH

**Table 5 medicina-57-01349-t005:** Estimation of the derivation of the STIMUL risk score model among the population of STEMI patients.

*n*	χ^2^	Degrees of Freedom	Log Likelihood	Cox and Snell’s R^2^	Nagelkerke’s R^2^	*p*
169	102.3	1	181.5	0.3	0.5	<0.001

**Table 6 medicina-57-01349-t006:** The simplified STIMUL risk score for predicting two-year mortality among the patient population with STEMI.

Independent Prognostic Factor	OR	95% CI		The Ratio of the Scope	Mortality Level %	Points
		Low	High			
Age, years						
˂40	–	–	–	–	< 1	0.5
40–59	1.8	1.2	1.9	1.6	21.0	1.0
60–69	2.1	1.2	2.7	2.3	26.0	2.5
≥70	3.7	2.1	4.8	2.3	52.0	3.5
Killip class						
0–I	–	–	–	–	–	0
≥II	15.5	8.6	28.1	3.3	82.0	0
Reperfusion						
yes	–	–	–	–	–	0
no	5.7	3.0	10.8	3.6	17.0	2.0
Resuscitated cardiac arrest	8.3	7.0	12.3	1.8	7.0	2.0
Diabetes mellitus	3.0	1.9	4.9	2.6	43.0	2.5
Signs of congestive heart failure prior to ACS	4.0	2.5	6.3	2.5	50.2	3.5

**Table 7 medicina-57-01349-t007:** The number of points for two-year death risk stratification in the simplified STIMUL risk score model among the patient population with STEMI.

Points	Risk %	Risk
**0–3.5**	5.0	LOW
**4.0–9.0**	40.0	MODERATE
**≥9.5**	70.0	HIGH

**Table 8 medicina-57-01349-t008:** Estimation of the derivation of the simplified STIMUL risk score model among the patient population following STEMI.

*N*	χ^2^	Log Likelihood	Cox and Snell’s R^2^	Nagelkerke’s R^2^	*p*
703	273.6	622.1	0.3	0.4	<0.001

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
