# Peer review of "Evaluation and Comparison of the STIMUL Extended and Simplified Risk Scores for Predicting Two-Year Death in Patients Following ST-Segment Elevation Myocardial Infarction"

_medicina, 2021, doi:10.3390/medicina57121349_

Round 1
Reviewer 1 Report
Korol et al. compared STIMUL risk scores with and without cardiac troponin levels in predicting 2-year mortality in patients with STEMI. The paper is generally well written and interesting. Here are the comments.
Major
- Abstract needs modification because the objective of the study and the results of cardiac troponin levels are written in Materials and Methods.
- In the abstract, the author wrote an addition of the troponin I increases the prognostic value by 10.7%. It should be clearly shown in the results.
- The c-statics for GRACE score with or without cardiac troponin levels should be also calculated and compared with STIMUL risk scores.
- Only 237 patients out of 1103 STEMI patients received primary PCI. The rate is much lower than the other continentals or countries. The generalizability to the other population might be doubtful. Moreover, the mean time from symptoms onset to admission at ED was relatively short (5.1hours), however, many patients did not receive primary PCI because of late arrival. The author should show more information about acute STEMI management in this cohort.
- Gender was associated with outcomes in this study. If gender is added to STIMUL risk scores, how does the results change?
Minor:
- The abbreviation of STIMUL should be shown in abstract.
- Continuous variables should be shown by mean (SD) or median (IQR). (e.g. time from symptoms onset to admission in results.)
- Reperfusion therapy in Table 1 might confuse readers because not receiving reperfusion therapy is associated with increased mortality.
- The author should calculate c-statistics to 2 decimal places.
- C-statistics for the simplified risk score is out of 95%-CI interval. Please correct it.
Author Response
Reviewer no 1.
Korol et al. compared STIMUL risk scores with and without cardiac troponin levels in predicting 2-year mortality in patients with STEMI. The paper is generally well written and interesting. Here are the comments.
Response: We would like to thank the reviewer for the opinion, the helpful comments, and the suggestions. We hope that the changes in the text of the manuscript are satisfactory and that you find the manuscript acceptable for publication.
Major
- Abstract needs modification because the objective of the study and the results of cardiac troponin levels are written in Materials and Methods.
Response: The abstract has been modified (lines 56-59).
- In the abstract, the author wrote an addition of the troponin I increases the prognostic value by 10.7%. It should be clearly shown in the results.
Response: This information was added to the Results section (lines 285-291).
- The c-statics for GRACE score with or without cardiac troponin levels should be also calculated and compared with STIMUL risk scores.
Response: Thank you for your suggestion.
The aim of the present study was to evaluate the 2-year death risk score in this specific population. The idea of comparing both of those scales can be performed in the future as well as validation of the STIMUL risk score in other populations. A reliable comparison of the GRACE and STIMUL scores requires the analysis of the 6-month risk score according to the GRACE score and a separate analysis of the 6-month risk score for the STIMUL population as GRACE estimates the probability of death within 6 months of hospital discharge. An estimation of the STIMUL 6-month risk score would be a separate issue (see PERSPECTIVES section).
- Only 237 patients out of 1103 STEMI patients received primary PCI. The rate is much lower than the other continentals or countries. The generalizability to the other population might be doubtful. Moreover, the mean time from symptoms onset to admission at ED was relatively short (5.1hours), however, many patients did not receive primary PCI because of late arrival. The author should show more information about acute STEMI management in this cohort.
Response: Specific comments were introduced in the Discussion section (lines 389-425)
As two of the Reviewers asked about the low rate of PCI in Ukraine, we would like to make a specific comment. Ukraine is a low-income European country ($3726 per capita income in 2020). At the time of the study (enrollment 2008–2010), there was only 1 Cath-lab in Khmelnitsky with 10 monitored beds in an Intensive Cardiac Care Unit (ICCU) and 1 Cath-lab in Vinnitsa with 8 beds in an ICCU (population of both cities with surrounding the towns and villages population exceeding 2 million), without 24/7 access for ACS.
In our study, the rate of reperfusion therapy in Ukrainian patients with STEMI was dramatically low, but it was comparable with the rate obtained by other authors. Moreover, there are studies which reported at this time a reperfusion rate of 0 [Kämpfer et al. 2017]. Even now, after the implementation of the “Stent for Life” Initiative, it is still worse than in Central and Western European countries. This disproportion not only concerns the rates of revascularization, but also other aspects in healthcare in patients following acute coronary syndromes – starting with reperfusion therapy, through to difficulties in accessing specialists, ending with low adherence to medical treatment. According to data from the follow-up of the STIMUL population, we also know that adherence to DAPT was 21.4% (55.6% in the population that underwent the PCI procedure) during the six-month follow-up period and 16.3% (39.2% in the population that underwent the PCI procedure) during the first 12 months after STEMI.
Low troponin assessment at admission to hospital were also due to economic reasons that are not taken into consideration in Central and Western European countries. During hospitalization in public hospitals, patients must pay for blood tests (testing and materials for blood sampling), medications administered during hospitalization, and even hospital bedding. At the time of the study, they also had to pay for stents. Most of them refuse to have any additional tests to reduce the cost of hospitalization or they refuse treatment altogether. Therefore, the rate of cardiac troponin measurement at admission was also low in this cohort.
- Gender was associated with outcomes in this study. If gender is added to STIMUL risk scores, how does the results change?
Response: Gender did not change the c-statistic significantly, and we observed that the death rate among men and women varied in different age groups. The average age among men and women who died in the following two years was 65.9 ± 1.4 versus 73.0 ± 1.1 years, respectively (OR 4.7; 95% CІ: 10,1- 4,1; Ñ€ <0.001). However, men who died in the following two years were 7 years younger than women. The rate of two-year death was 2.5 times higher among women – 30.7 versus 12.3% in men (Ñ€ < 0.001). For this reason, gender was excluded from the model as a separate prognostic factor (as in GRACE, mini-GRACE, PAMI, CADILLAC risk scores).
Prognostic value of age as predictor of two-year death in in men and women after STEMI
|
Factor |
Two-year death |
OR |
SD |
95% CІ |
Ð |
||
|
men |
women |
low |
high |
||||
|
Age, years |
65.9±1.4 |
73.0±1.1 |
4.7 |
1.5 |
-10.1 |
-4.1 |
<.001 |
Minor:
- The abbreviation of STIMUL should be shown in abstract.
Response: A change has been made. (Line 61)
- Continuous variables should be shown by mean (SD) or median (IQR). (e.g. time from symptoms onset to admission in results.)
Response: Lines (104, Table 1, 111-124, 129, 174, 183)
- Reperfusion therapy in Table 1 might confuse readers because not receiving reperfusion therapy is associated with increased mortality.
Response: A change has been made (now in Table 2)
- The author should calculate c-statistics to 2 decimal places.
Response: A change has been made.
C-statistics of extended score – C-statistic of 0.89 (95% CI 0.84–0.93; p < 0.001)
C-statistics of simplified score – C-statistic of 0.86 (95% CI 0.83–0.89; p < 0.001)
- C-statistics for the simplified risk score is out of 95%-CI interval. Please correct it.
Response: A change has been made. This occurred after rounding up the c-statistics. This is now correct after calculating the c-statistics to 2 decimal places.
Reviewer 2 Report
This is an interesting study in which authors sought to determine the utility of STIMUL score that is derived from their long-term registry of STEMI patients in predicting two-year death events after STEMI. One of the goals was to see how the risk prediction model will behave if cardiac troponin I is excluded or included in the multivariable model. They report that both models stratified patients into low, moderate, and high risks with a C-statistic of 0.9 while the addition of the level of troponin I increased prognostic value by 10.7%. Finally, they conclude that both STIMUL extended and simplified risk estimation models perform well in the prediction of two-year mortality risk following STEMI and this might be helpful in situations when cardiac troponins are not known by the physicians. I have the following concerns, as outlined below:
- Authors state that the time from symptom onset to admission at the emergency department was 5.1 hours +/- 0.3 hours while about 60% were admitted during the first six hours. What was your average door-to-balloon time?
- Only half of admitted STEMI patients underwent catheterization department while out of those only 237 received primary PCI. This is extremely unusual and low compared to other contemporary practices. The authors state that this was due to patients arriving late or catheterization laboratories being unavailable, however, this should be detailed. What was "too late" by the authors and what do they mean by cath lab not being unavailable?
- Furthermore, nearly 20% of reasons why PCI was not performed were contraindications (7.4%...very high), patient refusal (6.1%, also unusual), and uncertain diagnosis (6.3%). This is a rather large percentage of patients that did not undergo PCI and as such should be further explained.
- Please disregard phi and Cramer V coefficients for the analysis. Rather use a two-year risk that is calculated as a continuous variable and inspect continuous variables against it by Pearson r correlation analysis or, even better, multivariable-adjusted linear regression. Then only report p-values. Results under the Determination of risk score section are very busy and such analysis is redundant.
- Please make clear that the dependent outcome in your regression analysis was the event of death during the 2-year period (dead 0-no, 1-yes) and not "two-year mortality risk". This makes it much clear that your independent variables in Table 1. were tested as independent predictors for the event of death during the 2-year period. Please make this more clear throughout the text and mention it explicitly in the Methods section.
- What was the troponin assay (was that high-sensitivity assay or not)? Please define this.
- Not receiving reperfusion was a nearly 6-fold increase in mortality during the 2 years after STEMI. This is extremely important since it shows that this quality metric should be significantly improved at your institution to improve patient outcomes and enabling patients' access to PCI in your setting is a crucial healthcare initiative that should be presented to your authorities. It is an important implication for the discussion.
- Could you provide the absolute number and percentage of patients that were enrolled in STIMUL but died during the 2-year follow-up? This should be clearly stated in the results and also Kaplan-Meier survival curve would be welcome here.
- Please explain how did you do validation of your score? Did you enroll some other external population to validate your score? That is the only way how it should be done. If you ran your score against the same population that you derived it from initially this is not considered validation. Therefore, your study explored derivation of the score, but not validation. That should be done in external populations other than yours.
- Tables 5 and 6 are redundant, omit them. For the ROC curve, just state in the text what are the results (AUC, p-value, etc.) and keep the ROC curve.
- Cardiac troponins were measured in only 17.1% of STEMI patients which is staggering. However, this belongs more to the discussion, not results.
- Show Kaplan Meier curve showing survival of patients during 2-years by stratifying curves in low, intermediate, high-risk categories. This would be an important improvement of the manuscript.
- Authors should discuss the socioeconomic implications of their findings and their STEMI management because the utilization of primary PCI and troponin assays in this population was subpar and certainly does not reflect contemporary STEMI practice in most of the Western world. This has important ramifications on patient outcomes.
Author Response
Reviewer no. 2
We would like to thank the reviewer for the detailed opinion, the helpful comments, and the suggestions. We hope that the changes in the text of the manuscript are satisfactory and that you find the manuscript acceptable for publication.
This is an interesting study in which authors sought to determine the utility of STIMUL score that is derived from their long-term registry of STEMI patients in predicting two-year death events after STEMI. One of the goals was to see how the risk prediction model will behave if cardiac troponin I is excluded or included in the multivariable model. They report that both models stratified patients into low, moderate, and high risks with a C-statistic of 0.9 while the addition of the level of troponin I increased prognostic value by 10.7%. Finally, they conclude that both STIMUL extended and simplified risk estimation models perform well in the prediction of two-year mortality risk following STEMI and this might be helpful in situations when cardiac troponins are not known by the physicians. I have the following concerns, as outlined below:
- Authors state that the time from symptom onset to admission at the emergency department was 5.1 hours +/- 0.3 hours while about 60% were admitted during the first six hours. What was your average door-to-balloon time?
Response: In patients who underwent PCI, the door-to-balloon time was 4.5 ± 0.2 (1.2–7.8) hours, The timeframe for refundable PCI in Ukraine at this time was 6 hours from symptom onset in 2008–2009 and 12 hours in 2010.
- Only half of admitted STEMI patients underwent catheterization department while out of those only 237 received primary PCI. This is extremely unusual and low compared to other contemporary practices. The authors state that this was due to patients arriving late or catheterization laboratories being unavailable, however, this should be detailed. What was "too late" by the authors and what do they mean by cath lab not being unavailable?
Response: Specific comments were introduced in the Discussion section (lines 389-425).
As two of the Reviewers asked about the low rate of PCI in Ukraine, we would like to make a specific comment. Ukraine is a low-income European country ($3726 per capita income in 2020). At the time of study (enrollment 2008–2010), there was only 1 Cath-lab in Khmelnitsky with 10 monitored beds in an Intensive Cardiac Care Unit (ICCU) and 1 Cath-lab in Vinnitsa with 8 beds in an ICCU (population of both cities with the surrounding towns and villages population exceeding 2 million), without 24/7 access for ACS.
In our study, the rate of reperfusion therapy in Ukrainian patients with STEMI was dramatically low, but it was comparable with the rate obtained by other authors. Moreover, there are studies which reported at this time a reperfusion rate of 0 [Kämpfer et al. 2017]. Even now, after the implementation of the “Stent for Life” Initiative, it is still worse than in Central and Western European countries. This disproportion not only concerns the rates of revascularization, but also other aspects in healthcare in patients following acute coronary syndromes – starting with reperfusion therapy, through to difficulties in accessing specialists, ending with low adherence to medical treatment. According to data from the follow-up of the STIMUL population, we know that adherence to DAPT was 21.4% (55.6% in the population that underwent the PCI procedure) during the six-month follow-up period and 16.3% (39.2% in the population that underwent the PCI procedure) during the first 12 months after STEMI.
Low troponin assessment at admission to hospital in STEMI patients were also due to economic reasons that are not taken into any consideration in Central and Western European countries. During hospitalization in public hospitals, patients have to pay for blood tests, medications administered during hospitalization, and even hospital bedding. Most of them refuse to have any additional tests to reduce the cost of hospitalization. Therefore, the rate of cardiac troponins measurement was also low in this cohort.
Timeframe for refundable PCI in Ukraine at this time was 6 hours from symptoms onset in 2008–2009 and 12 hours in 2010.
- Furthermore, nearly 20% of reasons why PCI was not performed were contraindications (7.4%...very high), patient refusal (6.1%, also unusual), and uncertain diagnosis (6.3%). This is a rather large percentage of patients that did not undergo PCI and as such should be further explained.
Response: The registry was operationally in 2008–2013, patients were enrolled in 2008–2010 and were followed up during 24 months.
The higher prevalence of contraindications for reperfusion therapy (PCI but also fibrinolytic therapy) in the STIMUL registry was associated with a high bleeding risk, 10.61% (n = 117), and a very high bleeding risk, 5.23% (n = 58), as assessed by the CRUSADE score. Furthermore, the higher rates of patient refusal were related with economic reasons and low awareness of the disease.
- Please disregard phi and Cramer V coefficients for the analysis. Rather use a two-year risk that is calculated as a continuous variable and inspect continuous variables against it by Pearson r correlation analysis or, even better, multivariable-adjusted linear regression. Then only report p-values. Results under the Determination of risk score section are very busy and such analysis is redundant.
Response: Phi and Cramer V coefficients were removed from the analysis and the text. We changed the way of presenting the determination of risk score. However, at the moment, we are not able to provide new statistical analysis in the short timeframe for responding to the reviewers’ comments set by the editorial office. We do not have full access to the “raw data” as the first two authors have been temporarily posted to COVID-19 patient care.
- Please make clear that the dependent outcome in your regression analysis was the event of death during the 2-year period (dead 0-no, 1-yes) and not "two-year mortality risk". This makes it much clear that your independent variables in Table 1. were tested as independent predictors for the event of death during the 2-year period. Please make this more clear throughout the text and mention it explicitly in the Methods section.
Response: Changes were introduced to the text (and article title as well) in accordance with the reviewer’s suggestion.
- What was the troponin assay (was that high-sensitivity assay or not)? Please define this.
Response: It was the quantitative determination of cardiac specific troponins (Biomerica, USA) (Line: 115)
- Not receiving reperfusion was a nearly 6-fold increase in mortality during the 2 years after STEMI. This is extremely important since it shows that this quality metric should be significantly improved at your institution to improve patient outcomes and enabling patients' access to PCI in your setting is a crucial healthcare initiative that should be presented to your authorities. It is an important implication for the discussion.
Response: This issue was raised at the end of the Discussion section in accordance with the reviewer’s suggestions.
- Could you provide the absolute number and percentage of patients that were enrolled in STIMUL but died during the 2-year follow-up? This should be clearly stated in the results and also Kaplan-Meier survival curve would be welcome here.
Response: 169 individuals (19.38%) died within a two-year period after discharge (lines 138-141 + Fig. 1).
- Please explain how did you do validation of your score? Did you enroll some other external population to validate your score? That is the only way how it should be done. If you ran your score against the same population that you derived it from initially this is not considered validation. Therefore, your study explored derivation of the score, but not validation. That should be done in external populations other than yours.
Response: Thank you for this precious hint. As indicated by the reviewer, we performed a derivation of our score before the reviews. We have also pointed to lack of validation in other external population as a limitation of the study.
However, after the review corresponding author (Agnieszka Wsol MD) validated the STIMUL risk score with a Polish cohort of STEMI patients (n = 236) hospitalized in 2010–2012 at the Brodnowski Hospital in Warsaw from Dr Agnieszka Wsol’s database. In this cohort, the number of patients that underwent PCI was nearly 89.6% (n = 113) and the rate of reperfused artery was 85% (n = 107). Validation of the extended risk prediction model in a contemporary cohort was performed by evaluating its calibration (by the Hosmer-Lemeshow test) and the discriminatory capacity (the area under the ROC curve). The discriminative calibration of the STIMUL estimation model of two-year death after STEMI was lower but still acceptable with a c-statistic of 0.748 (95% CI: 0.698–0.791).
- Tables 5 and 6 are redundant, omit them. For the ROC curve, just state in the text what are the results (AUC, p-value, etc.) and keep the ROC curve.
Response: Changes has been made.
- Cardiac troponins were measured in only 17.1% of STEMI patients which is staggering. However, this belongs more to the discussion, not results.
Response: The rate of troponins at admission was low (17.1%). The overall troponin measurement during hospitalization was high – usually all patients had the estimation of CTNI performed during hospitalization. The low rate of troponin testing at admission was due to economic factors – one measurement of all parameters, including fasting glucose and lipids usually the next day after admission to hospital was more cost-effective for patients. The estimation of troponins at admission was a point of our interest as it is related with both the size of myocardial infarction and the delay in the first medical contact in patients with STEMI.
- Show Kaplan Meier curve showing survival of patients during 2-years by stratifying curves in low, intermediate, high-risk categories. This would be an important improvement of the manuscript.
Response: At the moment Fig. 1 was added and accorded to 3rd Reviewer suggestions we prepared Fig. 3, 5. For preparing Kaplan Meier curves showing survival of patients with a risk-stratification (low, intermediate and high) Authors need more time (1 week given for responses) to access “raw data”.
- Authors should discuss the socioeconomic implications of their findings and their STEMI management because the utilization of primary PCI and troponin assays in this population was subpar and certainly does not reflect contemporary STEMI practice in most of the Western world. This has important ramifications on patient outcomes.
Response: This issue was raised at the end of the Discussion section in accordance with the reviewer’s suggestions.
Reviewer 3 Report
Korol et al. compared extended and simplified STIMUL score for predicting two-year mortality in patients following ST-segment elevation myocardial infarction. The study is interesting but there are some points that require attention.
I wonder how many patients (percentage) with STEMI do not undergo troponin testing in this region?
If data is available, it would be beneficial if the authors could assess STIMUL utility in CV mortality, not just overall.
The authors may present baseline characteristics in the form of a table in order to make it more readable.
The authors should compare prognostic value between GRACE and simplified STIMUL, regardless of the fact that available data suggests that STIMUL and GRACE were of similar predictive value. In line with this it would be interesting to perform analysis in which STIMUL and GRACE without troponins would be compared.
I wonder why the authors stratified mortality risk in such a manner. In line with this, could the authors present (in form of a figure) histogram of distribution of their population according to the newly developed score.
In the discussion section the authors should provide future directions of their score model.
Author Response
Reviewer no 3
We would like to thank the reviewer for the detailed opinion, the helpful comments, and the suggestions. We hope that the changes in the text of the manuscript are satisfactory.
Korol et al. compared extended and simplified STIMUL score for predicting two-year mortality in patients following ST-segment elevation myocardial infarction. The study is interesting but there are some points that require attention.
COMMENT: I wonder how many patients (percentage) with STEMI do not undergo troponin testing in this region?
Response: The rate of troponins at admission was low. Overall troponin measurement during hospitalization was high – usually all patients had the estimation of CTNI performed during hospitalization. The low rate of troponin testing at admission was due to economic factors (one measurement of all parameters, including fasting glucose and lipids were usually performed the next day after admission to hospital). The estimation of troponins at admission was a point of our interest as it is related with both the size of myocardial infarction and the delay in first medical contact in patients with STEMI.
COMMENT: If data is available, it would be beneficial if the authors could assess STIMUL utility in CV mortality, not just overall.
Response: We would like to present only all-cause mortality. Information provided by the families of the patients that died is not a reliable source for determination of the cause of death for use in a publication. Our intention was to based on reliable data. We did not have an access to all death certificates.
Total 6-month death risk in STIMUL registry was 7,34% (n=64) with 87.5% (n=56) of supposed CV death. Total and supposed CV one year death risk in the Ukrainian registry was 16.6% (n=140) and 11.6% (n=101), respectively. Total 2-year mortality in STIMUL registry was 19.4% (n=169) and supposed CV death was 13.5 % (n= 118).
COMMENT: The authors may present baseline characteristics in the form of a table in order to make it more readable.
Response: A table was added in accordance with the reviewer’s suggestion.
COMMENT: The authors should compare prognostic value between GRACE and simplified STIMUL, regardless of the fact that available data suggests that STIMUL and GRACE were of similar predictive value. In line with this it would be interesting to perform analysis in which STIMUL and GRACE without troponins would be compared.
Response: The aim of the present study was to evaluate the 2-year death risk score in this specific population. The idea of comparing both of those scales can be performed in the future as well as validation of the STIMUL risk score in other populations. A reliable comparison of GRACE and STIMUL scores requires an analysis of the 6-month risk score according to the GRACE score and a separate analysis of the 6-month risk score for the STIMUL population as GRACE estimates the probability of death within 6 months of hospital discharge. An estimation of the STIMUL 6-month risk score would be a separate issue for future (see Perspectives section).
COMMENT: I wonder why the authors stratified mortality risk in such a manner. In line with this, could the authors present (in form of a figure) histogram of distribution of their population according to the newly developed score.
Response: Thank you for this suggestion. We introduced Figures into the Results section.
COMMENT: In the discussion section the authors should provide future directions of their score model.
Response: Future directions of the STIMUL score model were discussed in the “PERSPECTIVES” fragment.
Round 2
Reviewer 1 Report
The author well revised the manuscript.
I have no further comments.
Reviewer 2 Report
The authors have answered all my queries. No further questions.
Reviewer 3 Report
No further comments.